# Multiple Instances of Adaptive Evolution in Aquaporins of Amphibious Fishes

**DOI:** 10.3390/biology12060846

**Published:** 2023-06-12

**Authors:** Héctor Lorente-Martínez, Ainhoa Agorreta, Iker Irisarri, Rafael Zardoya, Scott V. Edwards, Diego San Mauro

**Affiliations:** 1Department of Biodiversity, Ecology, and Evolution, Faculty of Biological Sciences, Complutense University of Madrid, 28040 Madrid, Spain; ainhoaag@ucm.es (A.A.); dsanmaur@ucm.es (D.S.M.); 2Section Phylogenomics, Centre for Molecular Biodiversity Research, Leibniz Institute for the Analysis of Biodiversity Change, Museum of Nature Hamburg, 20146 Hamburg, Germany; i.irisarri@leibniz-lib.de; 3Departamento de Biodiversidad y Biología Evolutiva, Museo Nacional de Ciencias Naturales (MNCN-CSIC), 28006 Madrid, Spain; rafaz@mncn.csic.es; 4Department of Organismic and Evolutionary Biology, Harvard University, Cambridge, MA 02138, USA; sedwards@fas.harvard.edu

**Keywords:** aquaporin, amphibious fishes, adaptive evolution, emersion

## Abstract

**Simple Summary:**

The role of aquaporins (AQPs) in the adaptation of amphibious fishes to terrestrial environments was investigated using genome mining, phylogenetics, molecular evolution, and protein structure modelling. Evidence of adaptive evolution was found in 21 AQPs belonging to 5 different classes but predominantly to the AQP11 class. These sequence changes indicate that the modifications in molecular function and/or structure could be related to the process of adaptation to an amphibious lifestyle.

**Abstract:**

Aquaporins (AQPs) are a highly diverse family of transmembrane proteins involved in osmotic regulation that played an important role in the conquest of land by tetrapods. However, little is known about their possible implication in the acquisition of an amphibious lifestyle in actinopterygian fishes. Herein, we investigated the molecular evolution of AQPs in 22 amphibious actinopterygian fishes by assembling a comprehensive dataset that was used to (1) catalogue AQP paralog members and classes; (2) determine the gene family birth and death process; (3) test for positive selection in a phylogenetic framework; and (4) reconstruct structural protein models. We found evidence of adaptive evolution in 21 AQPs belonging to 5 different classes. Almost half of the tree branches and protein sites that were under positive selection were found in the AQP11 class. The detected sequence changes indicate modifications in molecular function and/or structure, which could be related to adaptation to an amphibious lifestyle. AQP11 orthologues appear to be the most promising candidates to have facilitated the processes of the water-to-land transition in amphibious fishes. Additionally, the signature of positive selection found in the AQP11b stem branch of the Gobiidae clade suggests a possible case of exaptation in this clade.

## 1. Introduction

Adaptation to new environments is challenging, but can also provide possibilities of increasing species diversification [1,2,3]. In particular, water-to-land transitions are among the most extreme habitat shifts in the history of life [4,5]. Compared to aquatic animals, those living on land must confront higher gravitational pressure and desiccation conditions. Consequently, emersion from water is a complex evolutionary process involving numerous morphological (biomechanical) and physiological (metabolic and biochemical) changes, which are mostly associated with locomotion, vision, audition, respiration, and desiccation [1,5,6]. Tetrapods (i.e., amphibians, reptiles (including birds), and mammals) arguably represent the most successful transition to life on land in vertebrates. Additionally, actinopterygian fishes also account for multiple independent cases of amphibious evolution that have occurred along their evolutionary history (reviewed in [1,7,8]). These events provide an excellent model system for studying and comparing the tempo and mode of complex adaptations that lead to terrestrialisation.

The so-called amphibious fishes typically inhabit intertidal areas, taking refuge in small pools during low tides and presenting several adaptations for emersion [1,7,9]. Many amphibious fishes are air-breathers [1], as is the case with mudskippers, which can gulp air [10], and killifishes, which use their skin as a gas exchanger [11,12,13]. Likewise, higher ammonia tolerance and the ability to actively excrete this compound appear to be widespread adaptations in amphibious fishes (e.g., [14,15,16,17]). There are also outstanding examples of terrestrial locomotion in otherwise amphibious lineages, such as those of the climbing perch (*Anabas testudineus*) and the walking catfish (*Clarias batrachus*) [18,19]. However, little is known about the molecular and physiological mechanisms underpinning water recovery, maintenance, and homeostasis during emersion.

Aquaporins or AQPs (earlier known as membrane intrinsic proteins (MIPs)) are transmembrane channels that carry water and small, uncharged solutes [20,21,22]. Their molecular structure is highly conserved and comprises six α-helices connected with five loops. These proteins tetramerise and form five pores in cell membranes (one in each monomer plus the central one) [23,24]. Two opposite NPA (Asn–Pro–Ala) motifs form the pore and bond with the water molecule, as well as determining which solutes can pass across the pore [25]. The aromatic arginine (ar/R) selectivity motifs filter solutes, and the differentially conserved amino acids in aquaglyceroporins (P1–P5) have been described so far as the most important motifs involved in solute selectivity [26,27,28]. Most of these amino acid residues map onto the external half of the aquaporin molecule, suggesting that this region is mainly involved in solute specificity. In contrast, most of the regulatory processes of the molecule occur on the cytoplasmic half of the protein (reviewed in [29]).

Besides water, AQPs can transport a plethora of compounds such as glycerol, urea, ammonia, CO_2_, reactive oxygen species (ROS), and hydrogen peroxide [30,31], suggesting a broad relevance in different physiological mechanisms. Up to 17 different vertebrate aquaporin subfamilies or classes have been described, which can be clustered into 4 main groups: (1) the aquaglyceroporins or GLPs; (2) the water-selective classical AQPs; (3) the unorthodox AQPs or superaquaporins; and (4) the AQP8-type or aqua-ammoniaporins [26,32,33]. AQPs are particularly abundant in the main organs for water recovery in fishes, including gills, intestines, and kidneys, and they have been broadly associated with osmoregulatory processes in fishes and with fish acclimation to different salinities (reviewed in [34]). In tetrapods, which acquired a terrestrial lifestyle arising from sarcopterygian fish ancestors, several AQPs have been involved directly in the mechanistic basis of water conservation (reviewed in [33]). Hence, it can be postulated that some AQPs could have been recruited to be involved in the physiological adaptation needed during the emersion of actinopterygian amphibious fishes as well. For instance, Ip et al. [35] postulated that the upregulation of an aquaporin in gills and skin of the climbing perch could be related to higher ammonia excretion.

Herein, the molecular evolution of AQPs in 22 teleost fish genomes was investigated in the context of water-to-land adaptations. This study extends our earlier work on the role of AQPs in the amphibious behaviour of mudskippers [36] and takes advantage of the recent availability of genomic data on additional amphibious fishes, thus providing a more comprehensive dataset and permitting more detailed and accurate comparative analyses. A robust phylogeny of AQPs was reconstructed based on the expanded dataset and was used as an evolutionary framework to catalogue AQPs into classes and paralogs, as well as to investigate molecular and adaptive evolution at the nucleotide sequence level. Our results uncovered numerous instances of adaptive evolution in different AQPs across the studied amphibious fish lineages, suggesting a crucial role of this protein family in the conquest of land.

## 2. Materials and Methods

### 2.1. Genome Mining and Phylogenetic Reconstruction

We built on our previous dataset [36], adding data from 18 new genomes of actinopterygian fish species that either exhibit a truly amphibious lifestyle or have undergone a degree of amphibiousness (reviewed in [1,7,8]). Comparable data were included from the genomes of 33 additional actinopterygian fishes (Appendix A). Since sarcopterygians are the sister group of actinopterygians, we employed the genomes of six tetrapods, along with the genome of the coelacanth, as well as four PCR-generated gene sequences of lungfishes (of the AQP0 and AQP2-like classes) as outgroups. Genome and isolate AQP sequence retrievals were performed following the protocol detailed in [37] from GenBank as of June 2021. In short, sequence similarity searches [38] using the BLASTX tool v2.2.28 were run locally to retrieve all sequence fragments that could be identified as AQPs with an E-value threshold of 1 × 10^−10^. When available, the BLASTP tool was run on protein files too, as double-check. Initial alignments at the nucleotide level were conducted using the L-INS-I algorithm in MAFFT v7.505 [39] to verify exon–intron boundaries [37]. Finally, gene sequences were translated into amino acids using Geneious Pro v9.1.8 [40], and a combined dataset was assembled and aligned as indicated above for subsequent phylogenetic analysis. All protein sequences used in the subsequent analyses were either of the entire gene or, if partial, of a minimum of 70 amino acids in length.

The best-fit site-homogeneous model of amino acid replacement (JTT [41] + Γ [42] + I [43]) was determined using the Bayesian information criterion (BIC) in ProtTest v3.4. [44]. The final AQP dataset was then subjected to maximum likelihood analysis (1) using IQ-TREE with 1000 ultrafast bootstrapping (UFBoot) and SH-aLRT pseudo-replicates each [45,46,47] and (2) with 1000 fast bootstrap replicates using the rapid hill-climbing algorithm of RAxML v8.2.10 [48]. The RAxML analysis was run on the CIPRES Science Gateway [49]. Phylogenetic tree figures were generated using iTOL v5 [50].

### 2.2. Estimation of Gene Family Evolutionary Histories

A reference species tree phylogeny was obtained from Timetree.org [51] and used to analyse gene family evolution. This reference tree was validated using the bony fish phylogeny established by Hughes et al. [52]. A few minor discrepancies only affected low-supported branches. The expansion and contraction of the AQP gene family across species were examined with the Bayesian Estimation of Gene Family Evolution (BEGFE) software [53] using default parameters and a single birth–death rate parameter (lambda, λ). With this method, we estimated the probability of gene family numbers expanding, contracting, or remaining constant on each node of the species tree. A total of 2 independent Markov chain Monte Carlo runs were conducted for 10 million steps, with sampling performed every 1000 steps. Finally, we assessed the convergence of the posterior distributions of the estimated parameters in Tracer [54].

### 2.3. Analyses of Adaptive Evolution

Individual CDS alignments for each of the AQP subfamilies present in our dataset were created and aligned using TranslatorX v1 [55] with the same MAFFT algorithm as above. Twelve subsets were created following the AQP classes recovered in Figure 1 and Figure 2 and Appendix A. This step was performed to maximise the number of phylogenetic informative positions and to reduce the number of gaps, which is key for the positive selection test that was employed. Highly partial gene sequences were removed, as well as alignment columns containing 5% or more gaps using Geneious Pro v9.1.8 [40]. Additionally, gene sequences were inspected manually, and highly variable regions due to poor alignment or low-quality sequences were removed. We derived the phylogenetic trees of these datasets using RAxML with the GTR [56] + Γ model of nucleotide substitution and the same settings as described above, and branch lengths were corrected as substitutions per codon. We decided to conduct phylogenetic analyses for each AQP subfamily in order to test whether these paralogs were able to reconstruct the reference species tree (i.e., their phylogenetic signal) or how much each departed from it. In a few cases, we manually modified poorly supported branches that likely resulted from stochastic error to standardise the topologies to the reference species tree and the bony fish classification of Hughes et al. [52].

Trees and alignments were analysed with the CODEML module of PAML v4.4 [57]. Tests of adaptive evolution were performed using the branch-site test 2 (null model A (MA) vs. model A) [58], which is currently among the most powerful approaches (see its strengths and caveats discussed in a very recent article by its own developer [59]). We calculated likelihood-ratio tests (LRTs) with null MA as the null hypothesis and MA as an alternative hypothesis and computed *p*-values using a mixed χ^2^ distribution, which was obtained by dividing by two the value of the χ^2^ distribution with one degree of freedom [57,60]. Due to the several tests conducted on the same tree topology, we calculated *q*-values (corrected *p*-values) for a False Discovery Rate (FDR) using the qvalue package of R [61]. For those tests wherein the LRT was significant, we calculated the posterior probabilities for site classes using the Bayes Empirical Bayes (BEB) [62] implemented in PAML, identifying the specific sites under selection on each branch. We tested for possible recombination events using GARD [63], as implemented in Datamonkey [64]. The results are shown in the Appendix A.

### 2.4. AQP 3D Structure Modelling

Protein 3D structures were predicted using multiple sequence alignments (MSAs) generated through an Mmseqs2 application interface as implemented in ColabFold [65], which uses the recently released AlphaFold2 source code [66]. Gene sequences were entered into the ColabFold notebook (https://colab.research.google.com/github/sokrypton/ColabFold/blob/main/beta/AlphaFold2_advanced.ipynb, accessed on 18 April 2023) with the following advanced features: msa_method = MMSeq2, num_models = 5, and num_relax = Top1. The quality of the best model was assessed using the mean local distance difference test (pLDDT). A pLDDT score of ≥60 was considered a reasonable model, and scores of > 80 indicated a very accurate model.

The 3D structures of human AQP10 (6F7H) and AQP1 (1H6I) were retrieved from the Research Collaboratory for Structural Bioinformatics Protein Data Bank (RCSB-PDB). Positively selected sites were superimposed onto these crystallographic 3D structures.

UCSF Chimera 1.15 was used to view and manipulate the molecular graphics (PDB files) for the modelled structures [67]. The effects of missense variants in protein structures were predicted using Missense3D [68].

## 3. Results and Discussion

### 3.1. Diversity of Amphibious Fish Aquaporins

Our genomic screening yielded the comprehensive protein alignment of 1006 gene sequences and 441 amino acid positions. This dataset was subjected to maximum likelihood analyses with IQ-TREE (−ln *L* = 172,474.585) and RAxML (−ln *L* = 172,803.455), both yielding highly similar topologies with slight differences that were mainly concentrated on low-supported branches (Figure 1 and Appendix A). The reconstructed trees recovered 16 AQP classes grouped into 4 main groups with strong statistical support: (1) the aquaglyceroporins (AQP3, 7, 9, and 10); (2) the water-selective classical AQPs (AQP0, 1, 2, 4, 5, 6, 14, and 15); (3) the unorthodox AQPs or superaquaporins (AQP11 and 12); and (4) the AQP8-type or aqua-ammoniaporins (AQP8 and 16; Figure 1 and Appendix A) [26,32,33]. Note that AQP13, a type of aquaglyceroporin that was originally described in platypus and Western clawed frog [33,69] (neither of which were included in the present work) was not found in any of the analysed genomes.

**Figure 1 biology-12-00846-f001:**
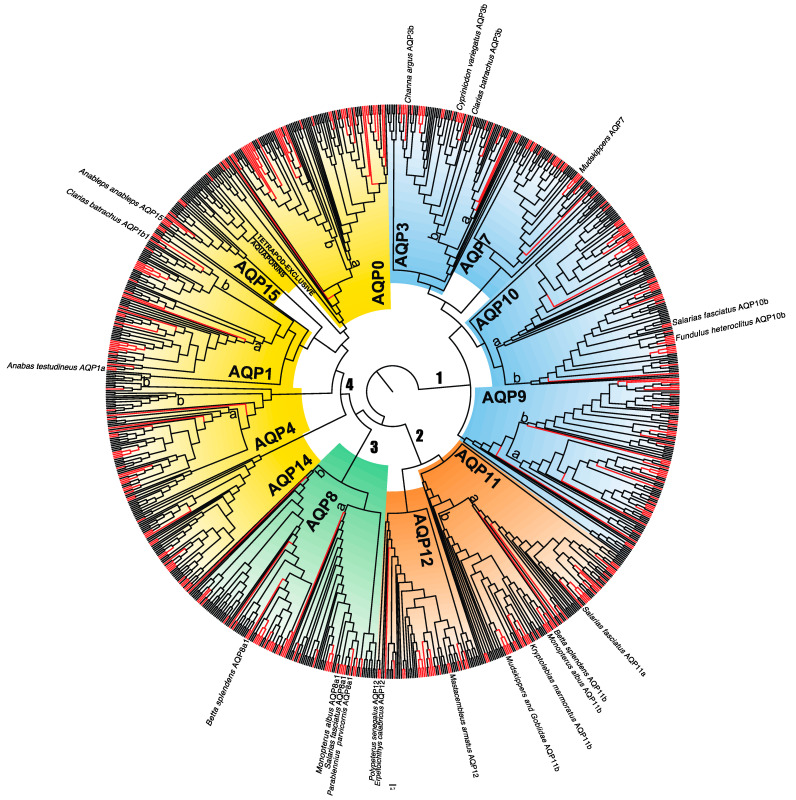
Maximum likelihood (IQ-TREE) cladogram of vertebrate AQPs based on 441 aligned amino acid positions. The tree was rooted using the split between aquaglyceroporins and the rest of the AQPs. The main four groups of AQPs are indicated with colour panels (blue for aquaglyceroporins, orange for superaquaporins, green for aqua-ammoniaporins, and yellow for water-selective classical AQPs). AQP paralog classes are named (AQP0–15), and paralog groupings within each class are denoted with letters (a, b, a1, and a2, following [33,69]) on the corresponding branches. Branches of amphibious fishes are highlighted in red. The names of the branches (species plus paralog) under adaptive selection are shown near their terminal locations on the tree. A detailed, fully labelled phylogram is available in Appendix A.

Phylogenetic relationships among AQP classes were generally unresolved due to low branch support (Appendix A). Based on the reconstructed phylogenetic patterns, many of the AQP paralogs were likely generated through two rounds of whole genome duplication (WGD) that occurred early in the evolution of vertebrates (dubbed R1 and R2, respectively) [33,69]. In addition, a more recent WGD event (R3) occurred on the stem branch of teleost fishes, providing them with a broader repertoire of AQP genes [70,71]. Furthermore, a fourth WGD event (R4) occurred in the common ancestor of salmonids and some cyprinids [72,73], thus acquiring even more copies of AQP genes, some of which were retained [74]. Conversely, the diversity of AQPs within actinopterygian fishes could also be explained by alternative events such as tandem or inter-chromosomal duplications. For instance, according to Finn et al. [75] a tandem duplication event on the branch that leads to all actinopterygian fishes AQP10, also yielded two copies in the non-teleost Reedfish (*Erpetoichthys calabaricus*) and Gray bichir *(Polypterus senegalus*).

Focusing on amphibious fishes, a total of 356 putative AQP genes were recovered (Figure 2), including 9 additional mudskipper AQPs that were not mined in our previous study [36]. These AQPs could be classified into 13 different classes (AQP2, 5, and 6 are exclusive of tetrapods). For most amphibious fish species, one paralog per class was found, no gene family expansions were detected, and only some putative cases of gene loss could be identified. This result should be interpreted with caution though, as the level of completeness, assembly, and annotation of genomes available in public databases is fairly uneven, and their quality is sometimes low, thus hampering proper gene isolation. In particular, few AQP15 orthologs were identified and several species appeared to lack this paralog (Figure 1 and Figure 2 and Appendix A). Finn et al. [33] reported that AQP15 orthologs were present prior to the emergence of all jawed vertebrates (Gnathostomata), but this was subsequently lost in many species, which was associated with a genome reduction event [76]. However, the potential physiological impact of this gene loss is not well understood. On the one hand, several studies have revealed examples of overlapping/redundant functions among different aquaporin classes (reviewed in [31]). Additionally, the patterns of expression of these genes are highly diverse, and even under similar physiological conditions, differences among species and organs can be found (see [34]).

**Figure 2 biology-12-00846-f002:**
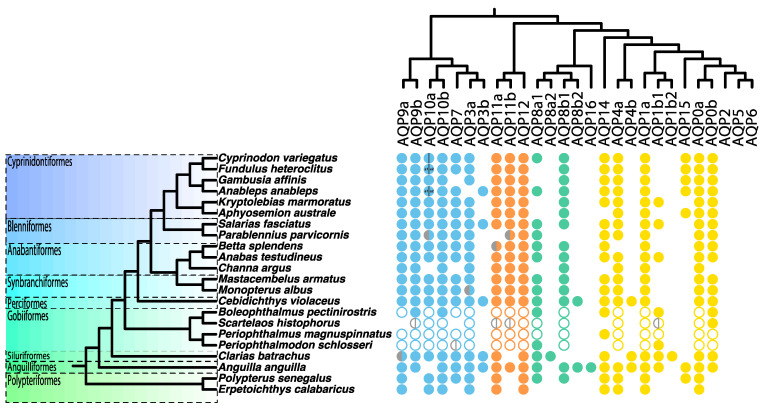
Aquaporin catalogue of the studied amphibious fishes. Filled circles denote complete gene sequences retrieved from whole-genome shotgun data in this study. Open circles denote AQPs retrieved from a previous study [36]. Half-grey circles denote partial gene sequences, i.e., gene sequences in which identification of the entire ORF was not possible. Blue circles correspond to the aquaglyceroporin group, orange circles represent superaquaporins, green circles indicate aqua-ammoniaporins, and yellow circles represent water-selective classical aquaporins. AQP1b1 and b2 paralog classifications are unclear. The existence of *Anguilla anguilla* AQP16 is unclear. The duplication of these classes (dubbed ‘a’ and ‘b’) mainly occurred on the branch of teleosts; therefore, *Erpetoichthys calabaricus* and *Polypterus senegalus* only possess one copy of each paralog, except for *E. calabaricus* AQP10 and *P. senegalus* AQP8 and 10.

To estimate the rates of AQP gene duplication and loss across vertebrates, we conducted a Bayesian analysis that estimated the birth–death rate, which was measured with a parameter called lambda. We assumed a model in which the lambda parameter is fixed across branches and obtained a value of 1.61 × 10^−3^. This value suggests a lower turnover within the AQP family compared to more variable families, such as the major histocompatibility complex (MHC) [77]. Furthermore, this test also allowed us to estimate the probability of expansion, contraction, or conservation of the AQP gene number in each group. For example, the branch leading to the Atlantic salmon (*Salmo salar*) was suggested to have undergone an expansion (Appendix A), a result that agrees with the R4 WGD event in this lineage [73].

Regarding the studied amphibious fishes, the northern snakehead (*Channa argus*) and the walking goby (*Scartelaos histophorus*) may have undergone a contraction of AQP genes. However, in the case of *S. histophorus*, the results could perhaps be related more to a poorly assembled genome (with lower-quality source data) than to a genuine contraction of AQP genes. In a similar way, our results suggest that the rock-pool blenny (*Parablennius parvicornis*) experienced a contraction of AQP genes, but again, the quality of this genome assembly was low and included several partial gene sequences (Figure 2). This low-quality assembly could have also misled the result found for the branch leading to the jewelled blenny (*Salarias fasciatus*), as the number of genes for *P. parvicornis* was very low. Surprisingly, our results suggest an expansion of the AQP repertoire in the European eel (*Anguilla anguilla*) (Appendix A). This could not only be due to the retention of several AQP paralogs, such as AQP4b or AQP15, but also the presence of a putative AQP16 (Figure 2). Finally, the Philippine catfish (*Clarias batrachus*) AQP repertoire seems to have remained unchanged with respect to the ancestor.

Altogether, these results indicate a complex evolutionary pattern that extends beyond birth and death gene family processes and cannot be fully understood using presence–absence approaches alone. However, the absence of exclusive copies in the studied amphibious fishes suggests that if AQPs played a role in achieving an amphibious lifestyle in any of these species, this could be related to selective changes in gene sequences that are present in their fully aquatic sister groups as well [78].

### 3.2. Adaptive Evolution in Amphibious Fish AQPs

The footprints of positive selection can be detected in gene sequences by estimating the ratio between non-synonymous and synonymous nucleotide substitutions (*d*_N_/*d*_S_), usually known as the selection coefficient or omega (ω) [79]. Apart from our earlier study on mudskippers [36], two other recent studies have related positive selection in AQPs to water habitat change in vertebrates: one in squamates during their adaptation to life in dry habitats [80,81], and the other focused on the cetacean land-to-water transition [80,81]. Similarly, branch-site tests were conducted to search for positions under positive selection in each paralog in those branches of the reconstructed tree that led to amphibious fish species and thus could be related to adaptation for the water-to-land transition.

A total of 21 branches of amphibious fishes, which expand across 7 different orders of Actinopterygii, showed footprints of adaptive selection in AQPs (Table 1 and Figure 1). Of these 21 branches that may have undergone adaptive evolution, 8 of them clustered within the superaquaporin group (AQP11 and 12) (Table 1 and Figure 1) [26,33]. Orthologs of both of these classes have been shown to be upregulated in seawater in one marine medaka [82] but downregulated in the roughskin sculpin [83], indicating a potential role in osmoregulation. Moreover, AQP11 has been found to transport hydrogen peroxide (H_2_O_2_) and is associated with cellular stress reduction in the endoplasmic reticulum [30,84,85]. This compound can also be transported by the AQP3, AQP8, and AQP9 proteins, suggesting that the role of AQPs in the ROS pathway could be more important than previously thought [86,87,88]. However, it remains unknown how these proteins cope with the increase in ROS and oxidative stress during the adaptation of fishes to land and air-breathing conditions, as well as their specific functions and expression patterns.

Adaptive evolution was detected on four branches within the aqua-ammoniaporins, three of them corresponding to the same paralog, AQP8a1 (Table 1 and Figure 1). These proteins are the main AQPs that are able to transport ammonia; therefore, they have been strongly associated with excretion and detoxification [89]. Our results suggest that adaptive evolution could have occurred on the branch leading to the mudskippers clade, the swamp eel (*Monopterus albus*), the Siamese fighting fish (*Betta splendens*), and *P. parvicornis*. Among the mudskippers, there are some species that are capable of excreting ammonia through gills during emersion [15,90]. There is also evidence of ammonia detoxification to glutamine during emersion in *M. albus* [14,91,92]. However, there is still no evidence of ammonia excretion in terrestrial conditions in either *B. splendens* or *P. parvicornis*. Ammonia transport is not restricted to this AQP, and there is evidence of ammonia transport in the AQP1, 6, and 9 orthologs (reviewed by [31]). One study suggested a role of an AQP1 ortholog in ammonia excretion in *A. testudineus* during emersion [35]. Nevertheless, even though our dataset included sequence data for AQP1, we did not find any signature of adaptive evolution in this gene.

Up to six branches were found to have undergone positive selection within the large clade of GLPs, with three of them clustering within the AQP3 class. There is evidence of the downregulation of an AQP3 ortholog in *F. heteroclitus* embryos during aerial exposure, likely to reduce water loss [93]. However, as in *A. testudineus*, no signal of adaptive evolution was found in the *F. heteroclitus* AQP3 branches. Therefore, the relationship between our results and the evolution of an amphibious lifestyle in these fishes remains unclear.

We identified adaptive evolution in the AQP10b of *S. fasciatus* (Table 1 and Figure 1), which, like *P. parvicornis*, belongs to the Blennidae family known for its notable amphibious behaviour [7,94]. For example, the Kirk’s blenny (*Alticus kirki*), another member of this group, exhibits a pattern of higher urea excretion during both emersion and immersion [95], whereas the shanny (*Blennius pholis*) can volatilise ammonia through its skin [96]. Notably, although instances of adaptive evolution in the aquaporin of these blennies are scarce, both occurred in AQPs involved in urea (AQP10 [89,97]) and ammonia (AQP8) transport.

Finally, only three branches in the water-selective classical AQPs clade (Figure 1) were found to have potentially undergone adaptive evolution, with two of them likely being unreliable, as indicated below, and the remaining one not showing any signal of adaptive evolution. Despite initially being considered as merely water channels, it is now known that classical AQPs can transport a wide variety of solutes (reviewed in [31]). It is worth noting that tetrapods, which successfully transitioned to an amphibious and later fully terrestrial lifestyle, rely on the emergence of three novel paralogs (AQP2, 5, and 6 [33]) that belong to the clade of classical AQPs. In this study, we initially hypothesised a possible convergence hallmark within the AQP family between tetrapods and actinopterygian amphibious fishes. However, this was not the case according to our results. These results suggest that if AQPs contributed to the evolution of amphibious lifestyles in actinopterygian fishes, this may have occurred through molecular changes at the sequence level that are very dissimilar to those relevant to the water-to-land transition of tetrapods.

Despite having signatures of adaptive selection in 21 branches of amphibious fishes, specific positions under positive selection could only be identified in 12 out of the 21 branches (Figure 3). This discrepancy could indicate that in some branches, the signal of adaptive evolution was cumulative and not strong enough at any particular site. Moreover, the branch-site test is generally considered conservative and sometimes may lack enough statistical power [59,62] (see also [98,99]). Another potential caveat may be the presence of highly variable regions in some AQPs (which could reflect fast evolutionary rates or poor sequence quality), as the employed tests heavily rely on robust alignments [58]. Therefore, positive selection analysis should be interpreted with caution, especially considering the uneven (sometimes low) quality of genome assemblies available in public databases. In this regard, the results obtained from the AQP10b branch of the mummichog (*Fundulus heteroclitus*) and the AQP1b1 of *C. bactrachus* (Table 1 and Figure 1) may be questionable, because positively selected sites were found in highly variable regions, possibly due to low-quality gene sequences. To avoid misleading results, we discarded both results from further analysis. Finally, a seemingly contradictory result was found on the branch leading to largescale four-eyes (*Anableps anableps*) for AQP15 (Table 1). The branch-site test suggested a statistically significant event of adaptive selection, but the associated ω-value indicated neutral evolution (ω = 1). This incongruence may be directly related to the small number of AQP15 orthologs in the analysed dataset, as branch-site tests might not perform well in such cases [58]. Additionally, the branch-site test can also be affected by recombination events, especially when they occur at a high frequency, such as in viruses [100]. When recombination occurs, phylogenetic inference can be misled, and the *d*_N_/*d*_S_ ratio can be inflated, providing false positives. In order to deal with this problem, we examined recombination for each of the individual (subfamily-level) alignments that were used for the positive selection analyses using GARD [63]. We only found evidence of possible recombination in the AQP12 and AQP15 datasets (Appendix A), thus suggesting that such results should be interpreted with slightly more caution. As discussed below, the AQP15 dataset is too small to be considered reliable for positive selection analyses. On the other hand, the results depicted in Table 1 show that some ω-values are very high (equal or similar to 999). These values suggest a very small or even null *d*_S_ value but can be interpreted as artefactual estimates. All these very high values were indeed found on the branches reported as possibly unreliable (see above) or in those where no specific sites were highlighted as positively selected (Table 1). Thus, such results should be interpreted with caution.

### 3.3. Adaptive Evolution in Gobioidei AQP11b

The branch-site test detected signatures of adaptive evolution in mudskipper AQP11b, AQP8a1, and AQP7 but could not assign them to any particular amino acid (Table 1 and Figure 3). In contrast, our previous study [36] identified several positively selected sites in mudskipper AQP10a and AQP11b. The main differences between both studies are the increased taxon sampling herein and a more thorough genome mining pipeline [37], which helped us find orthologs and paralogs that were previously overlooked (Figure 2). Regardless, AQP11b emerged in both studies as a promising candidate gene undergoing adaptive evolution. In the previous study [36], one site under selection was particularly interesting because it was located in the HH1 NPA motif of AQP11b, which was substituted with an SFI (Ser-Phe-Ile) motif. The two opposite NPA motifs form the pore of the AQP and bind with the water molecule, as well as determining which solutes can pass across the pore [25]. Herein, adaptive evolution in the same NPA motif of AQP11b was detected, but on the Gobiidae stem branch (Table 1 and Figure 3). This suggests that the modification of the NPA motif predated the origin of the mudskipper clade, likely being an apomorphy of the entire Gobiidae or even Gobioidei. Since not all Gobioidei have an amphibious lifestyle, it is still unclear whether the modification of the NPA motif could be associated with a potential advantage in favouring water recovery during emersion in mudskippers. If this were the case, it might represent a case of exaptation [101].

Recently, the genome sequences of additional Gobioidei have been published (plus one new Apogonoidei, *Siphamia tubifer*), but they could not be included in our analytical batches because they were released after we completed our primary data acquisition. These new gobioid genomes correspond to three additional representatives of Gobiidae (*Mugilogobius chulae*, *Proterorhinus semilunaris*, and *Rhinogobius similis*), one Butidae (*Bostrychus sinensis*), and two Odontobutidae (*Neodontobutis hainanensis* and *Perccottus glenii*). We searched and extracted the AQP11b gene sequence from each of these new genomic data (as detailed above) and aligned them with those included in our analyses in order to specifically compare the HH1 NPA motif region. The serine found to be under adaptive selection (compared with asparagine in Apogonoidei) is present in all Gobioid taxa (Appendix A), confirming that this change in the NPA motif is, in fact, apomorphic to the entire Gobioidei.

Given that several early-branching gobioid lineages (Odontobutidae and Rhyacichthyidae, Milyeringidae, Eleotridae, and Butidae) are mainly freshwater (or brackish) compared with nearly fully marine Apogonidae, Kurtidae, and Trichonotidae (closer relatives of Gobioidei), it can be postulated that gobioid ancestors likely underwent a transition from marine to freshwater environments [102,103]. In this context, the adaptive change detected in the AQP11b of gobioids could indeed be related to their marine-to-freshwater transition (which involves major osmoregulatory adjustments). Phylogenetically more derived, the mudskipper stem branch of AQP11b also shows an independent signal of adaptive selection (Table 1), as did some specific branches of AQP11b within the mudskipper clade itself in our earlier study [36]. This might suggest that the NPA motif change possibly related to freshwater adaptation in Gobioidei ancestors could have been later exapted in mudskippers during their transition to terrestrial environments. Interestingly, a similar pattern of facilitation of terrestrial adaptation through freshwater/brackish intermediate steps has been recently reported for invertebrate AQPs as well [78].

### 3.4. Mapping of Positively Selected Sites onto the 3D Structure

Given the importance of NPA motifs (see above), conformational changes in or around them could influence solute selectivity by altering the pore structure. We found evidence of positive selection in an amino acid adjacent to the first NPA motif as well as an amino acid adjacent to the first ar/R filter on the AQP11b branch of *B. splendens* (Figure 3 and Figure 4). Additionally, from the aforementioned result on the Gobiidae stem clade, these two positions are closer to some of the most important AQP motifs. Two other positively selected sites were located on the external half of the molecule (*S. fasciatus* AQP10b and 11a; Appendix A) along with the NPA and the ar/R filters, suggesting a possible role in solute recognition as well.

Our results show that most amino acids that experienced positive selection are located in transmembrane regions, which are typically better conserved than the connecting loop regions (Figure 3 and Appendix A). Only six positions were found on the N- and C-terminal regions. There is evidence of amino acids in these regions that are related to AQP regulation, including gating and trafficking (reviewed in [104]). However, the mechanism is not well characterised and can vary among AQP paralogs. We did find a few positions on some AQP3 and AQP12 orthologs that are located on the C-terminal part of the protein that could be related to changes in regulation. However, these results should be interpreted with caution because these regions are much more variable than the rest of the protein. Additionally, in the case of AQP12, there is some evidence of possible recombination, and this could have influenced the result (see above). Finally, there are three amino acids that are shared between more than one lineage. One position, close to the fourth amino acid site that is reported to confer glycerol selectivity in aquaglyceroporins, is shared between gobies, the mangrove rivulus (*Kryptolebias marmoratus*), and *B. splendens* in the AQP11b ortholog (Figure 3 and Figure 4). Similarly, *M. albus* and *P. parvicornis* share a position in an AQP8 ortholog. Moreover, gobies and *M. albus* share amino acids at the same position between an AQP11 and an AQP3 ortholog. Our knowledge of the scope of these results is still limited; however, their signals are noteworthy and may constitute a case of convergent evolution.

Altogether, these results suggest that point mutations under positive selection could have modified AQP function and regulation as well as solute permeability during the evolutionary history of actinopterygian fishes and could perhaps have facilitated the convergent transition to an amphibious lifestyle. However, how these substitutions may have modified the molecular structures of these proteins and thus, their functions, especially the one detected in the first NPA motif of the Gobiidae clade, needs further research and a better understanding of the 3D structure of every AQP class.

To test the impact of the modification of the NPA on the AQP11b stem branch of the Gobiidae clade, we employed AlphaFold2 [66] to model the AQP11b of *Periophthalmus magnuspinnatus* and *Neogobious melanostomus* as representatives of the Gobiidae family, using the zebrafish (*Danio rerio*) AQP11a as a model organism and the orbiculate cardinalfish (*Spaheramia orbicularis*) AQP11b as the closest sister species with a canonical NPA motif. Our results show a high-quality reconstruction for all the modelled proteins (pLDDT > 80), with the exception of some inaccurately modelled tails (Appendix A). However, upon mapping these structures with human AQP1 (*Homo sapiens*), we observed seven alpha helices instead of the canonical six. To determine if this additional helix was unique to AQP11 orthologs, we also modelled an AQP1 ortholog of *D. rerio* and the human AQP1 ortholog from the RefSeq database at the NCBI [105]. Our results show that the AQP1 ortholog of *D. rerio* folds similarly to the human AQP1 and only six alpha helices are present (Appendix A). Therefore, this suggests that maybe AQP11 orthologs could fold another alpha helix. However, further crystallographic studies are needed to accurately determine the protein structures of AQP11 orthologs and their implications in functions. Nevertheless, this does not invalidate the accuracy of the modelling in the highly conserved regions of the AQP molecule (Appendix A). Notably, the region where the modified NPA (SFI) motif of Gobiidae is located exhibited the largest differences among the modelled proteins (Appendix A). Missense·3D [68] was used to calculate the impact of the substitution of an asparagine (N) for a serine (S) on the structure of *D. rerio* and *S. orbicularis* AQP11. The program predicted that this mutation alters the cavity and leads to the contraction of the cavity volume in both proteins, a reduction in 123.552 Å3 and 104.112 Å3, respectively. This result suggests a possible modification of the pore that could result in a modification of its function, although functional analyses are needed to characterise these mutations.

Few studies have focused on how AQPs function during emersion in actinopterygian fishes. In this study, we have shown that several cases of adaptive evolution have occurred in these gene sequences during the evolution of amphibious fishes. We have focused on the molecular evolution of AQPs regarding coding regions and testing adaptive selection that can be related to the emersion of these fishes. However, some amphibious fish species did not exhibit any signal of positive selection (Table 1 and Appendix A), and this may be related to the different degrees of amphibiousness considered. For example, we did not find any branch under selection in either the mosquitofish (*Gambusia affinis*), a species that can be considered fully aquatic although it can leave water for predator avoidance for a few minutes, or in fishes that are able to persist for hours out of water, such as the European eel (*A. Anguilla*) [1]. This suggests a more complex evolution regarding AQPs and the different instances of amphibious behaviour, all of which need further research. Moreover, several other aspects such as patterns of expression or aquaporin regulation were out of the scope of this study, but they could certainly be equally if not more relevant than protein evolution to the water-to-land transitions of these amphibious fishes [106].

## 4. Conclusions

The multiple instances of positive selection that may have modified the molecular functions and/or structures of the AQPs in amphibious fishes suggest the possible role of these proteins in adaptation during water-to-land transitions. In particular, AQP11 orthologs are the most promising candidates for further investigation within the amphibious fish framework because almost half of the reported tree branches and amino acid positions under positive selection correspond to gene sequences of this AQP class. The positively selected sites in the HH1 NPA motif of AQP11b in gobioids could represent a case of exaptation. Herein, a robust phylogenetic framework was reconstructed to provide thorough bioinformatic cataloguing of AQP paralogs and to detect instances of adaptive evolution. Further confirmation of this hypothesis and the role of detected sequence changes under selection in the adaptation to terrestrial conditions in these amphibious fishes requires integrating our findings with physiological, biochemical, and even ecological insights.

## Figures and Tables

**Figure 3 biology-12-00846-f003:**
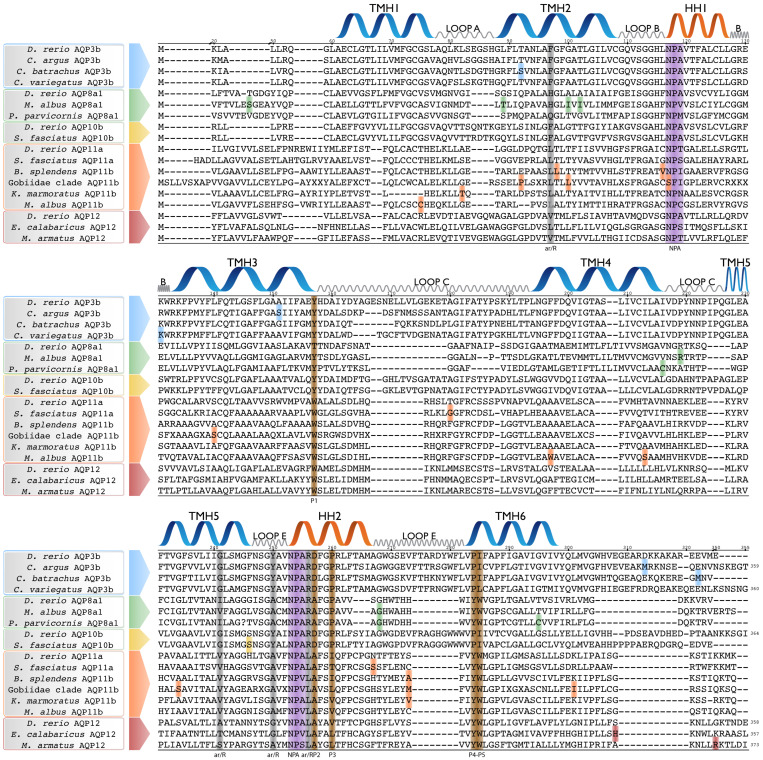
Sequence alignment and structural annotation of positively selected AQPs. Only those amphibious fish AQPs with sites under positive selection are shown. Colour highlights denote sites under positive selection as follows: blue for AQP3, green for AQP8, yellow for AQP10, orange for AQP11, and red for AQP12. Gene sequences of the corresponding paralogs of *Danio rerio* are included as references. The transmembrane helix (TM1-6; blue), the hemi-helices (HH1-HH2; orange), and loops A–E (grey) are annotated for *D. rerio* Aqp10b based on a molecular sequence wrap of the crystallographically resolved structure. NPA motifs (purple), ar/R selectivity filters (grey), and sites reported to confer glycerol selectivity (P1–P5) (blue) are highlighted following [26].

**Figure 4 biology-12-00846-f004:**
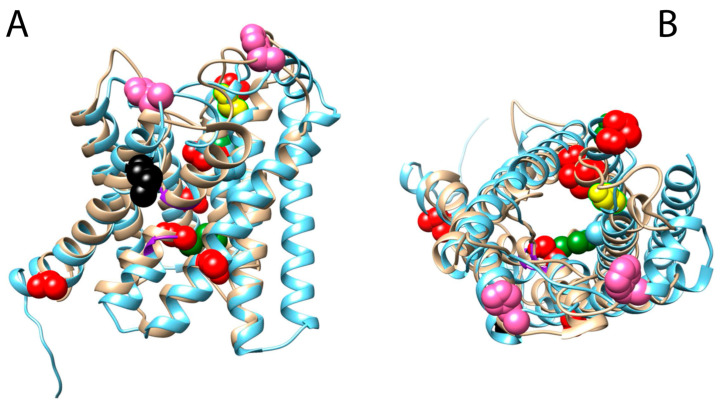
Three-dimensional structural model reconstruction of the *Periophthalmus magnuspinnatus* AQP11b gene. Superimposition of the AlphaFold2 model (blue) with human AQP1 (PDB ID: 1H6I) (beige). NPA boxes are highlighted in purple. (**A**) Lateral view. (**B**) Top view from outside of the cell membrane. Blue dots correspond to *Monopterus albus* AQP11b, green to *Betta splendens* AQP11b, pink to *Salarias fasciatus* AQP11a, red to the Gobiidae clade AQP11b, and yellow to *Kryptolebias marmoratus* AQP11b. Black dots correspond to a shared position among *B. splendens*, *K. marmoratus*, and the Gobiidae clade AQP11b branches.

**Table 1 biology-12-00846-t001:** Results of the branch-site tests that were significant (*q*-value of the LRT < 0.05). All other tests were not significant.

AQP	Foreground Branch	LRT	*p*-Value ^a^	*q*-Value ^b^	ω ^c^	Prop. 2a ^d^	Prop. 2b ^e^	Selec. Sites
1a	*A. anableps*	7.711	0.003	0.0298	999	0.006	0.001	0
1b1	*C. batrachus **	21.708	1.59 × 10^−6^	1.126 × 10^−4^	136.799	0.071	0.007	1
3b	*C. argus*	16.551	2.367 × 10^−5^	2.603 × 10^−4^	31.599	0.032	0.008	2
3b	*C. batrachus*	19.478	5.087 × 10^−6^	1.119 × 10^−4^	49.655	0.037	0.009	3
3b	*C. variegatus*	9.906	8.233 × 10^−4^	0.006	67.061	0.013	0.003	1
7	Mudskipper clade stem	8.639	0.002	0.026	999	0.015	0.004	0
8a1	*M. albus*	12.334	2.217 × 10^−4^	0.007	12.806	0.068	0.013	6
8a1	Mudskipper clade stem	7.295	0.003	0.027	36.256	0.019	0.004	0
8a1	*P. parvicornis*	7.997	0.002	0.028	23.468	0.085	0.017	3
8b1	*B. splendens*	7.460	0.003	0.028	8.669	0.039	0.008	0
10b	*F. heteroclitus* *	47.896	2.247 × 10^−12^	8.76 × 10^−11^	67.772	0.005	0.001	1
10b	*S. pavo*	13.001	1.557 × 10^−4^	0.003	59.219	0.019	0.003	1
11b	*Betta*	15.126	5.029 × 10^−5^	8.298 × 10^−4^	43.954	0.030	0.007	3
11b	*Kryptolebias*	6.510	0.005	0.029	38.602	0.022	0.005	2
11b	*Monopterus*	13.291	1.334 × 10^−4^	0.001	18.128	0.038	0.009	3
11b	Mudskipper clade stem	9.607	9.690 × 10^−4^	0.006	999	0.029	0.006	0
11a	*S. fasciatus*	12.613	1.916 × 10^−4^	0.002	16.503	0.047	0.011	2
12	*E. calabaricus*	10.325	6.560 × 10^−4^	0.004	708.322	0.021	0.005	1
12	*M. armatus*	10.907	4.790 × 10^−4^	0.004	998.999	0.014	0.003	1
12	*P. senegalus*	11.961	2.717 × 10^−4^	0.004	999	0.016	0.004	0
15	*A. anableps*	9.594	9.761 × 10^−4^	0.008	1	0.057	0.017	0
11b	Gobiidae stem branch **	17.813	1.218 × 10^−5^	4.021 × 10^−4^	41.228	0.110	0.025	7

^a^ Uncorrected *p*-value of the LRT. ^b^ Multiple-test correction of the LRT *p*-value (false discovery rate). ^c^ Omega (*d*_N_/*d*_S_) ratio of the foreground branch(es). ^d^ Proportion of sites that are under positive selection (ω 2a > 1) on the foreground branch(es) and under negative selection (ω < 1) on the background branches. ^e^ Proportion of sites that are under positive selection (ω 2a > 1) on the foreground branch(es) and under neutral selection (ω = 1) on the background branches. * Non-reliable results (see Section 3). ** Not a fully amphibious clade.

## Data Availability

This study did not generate any new sequence data but used data already available in the NCBI Genome Database (https://www.ncbi.nlm.nih.gov/genome/ (accessed on 21 June 2021)). The sequence alignments and phylogenetic trees used for the PAML analyses are available in the Appendix A.

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
