# Peer review of "Multiple Instances of Adaptive Evolution in Aquaporins of Amphibious Fishes"

_biology, 2023, doi:10.3390/biology12060846_

Round 1

Reviewer 1 Report

This is an interesting contribution that explores the role of aquaporins proteins during the water-to-land transition in amphibious teleost fishes. The authors, first, mined genomic databases to retrieve and curate a dataset of aquaporine proteins, and performed phylogenetic analyses to delimit the orthology/paralogy of the different groups of proteins. They then performed selection analyses to detect branches along the phylogenetic trees, and specific aminoacidic positions, under positive selection. Finally, they associated the positive selected sites to potential changes in the protein structures and, hence, their functionality. They interpret all their results in terms of the potential selection of these structural/functional changes along the water-to-land transition in this group of amphibious fishes.

The manuscript is clearly organized and well written. Figures are illustrative and well presented. The methods and results are scientifically sound. In general, the manuscript constitutes a robust contribution that might be of interest to Biology readers.

I have only a few minor (editing) comments and a couple of general methodological questions or concerns:

11-      The authors obtained a reference species tree from a public database to analyze gene family evolution. On the other hand, to analyze patterns of selection for each protein, the authors gathered individual alignments for these proteins, and performed phylogenetic analyses. In some cases, after the analyses, they manually modified the resulting topologies to match known relationships among taxa. Wouldn’t make sense to use the reference species tree for the selection analyses too?

22-      It is well known that recombination among orthologues/paralogues in complex proteins families may increase false positives in selection analyses. Have the authors considered this possibility?

Minor comments:

-          Line 90: delete “is investigated”.

-          Line 135: Strictly speaking, “convergence of posterior distributions of the estimated parameters” can be assessed in Tracer if several runs are performed, but not with a single run as in this study. Convergence and mixing of the MCMC analyses should be explicitly tested in any bayesian analysis.

-          Line 151 (and reference list): update reference information.

-          Line 337: rewrite this sentence.

Reviewer 2 Report

The manuscript by Lorente-Martinez and coworkers deals with the molecular evolution of aquaporins (AQPs) and their possible role in the transition from water to land in the multiple occurrences of amphibious fishes across teleosts. The topic is interesting, the study is generally well-designed, the figures and tables are nice and the main text is overall pleasant to read.

As a "pipeline" phylogeneticist, I am less comfortable with some methodological choices (especially some key manual steps) of the authors, which (in my view) prevent perfect reproducibility. However, I do not think that these concerns would warrant additional analyses. Thus, simply elaborating in the text to better explain or justify some decisions would be enough for "Biology".

Below, I try to group my comments by topic, mostly following the line numbering. They are all meant to improve the final manuscript and can mostly be considered as minor points and/or caveats that would deserve to be mentioned in the text.

- Introduction

1. When discussing water-to-land transition, one immediately thinks about tetrapod evolution. The present work does not deal with that issue, but since it is only evacuated late in the introduction (lines 81-84), the reader might be confused to read about actinopterygian fishes going to land (lines 46-48). It might be better to mention early the sarcopterygian fishes and tetrapods (as in the abstract) to highlight the fact that the focus of the study is about the similar but less "successful" transitions observed in multiple lineages of actinopterygian fishes.

2. I would not conflate actinopterygian fishes (lines 24, 86) and teleosts (lines 47, 89, 106, 192, 284). Choose one of the two clades.

3. line 40: I am not sure about the meaning of "abrupt" in this context. This word suggests a rapid transition, but it does not seem required here. Maybe prefer "dramatic" or even "sharp"?

- Methods: Dataset assembly

4. lines 105-108: I don't exactly understand why sarcopterygian fishes and tetrapods were used as outgroups, as they are clearly derived, except maybe for functional annotation purposes. I would have expected to see sharks or lampreys as outgroups in such a study. For example, in the results (lines 251-253), the origin of AQP15 cannot be addressed because of this choice.

5. lines 108-110: "genome and isolate AQP sequence retrievals" is unclear. Did you start only from complete genome sequence data or did you also include isolate sequences (e.g. PCR-amplified) retrieved from GenBank? If so, this might create uneven completeness of gene catalogs and/or lead to redundancy issues.

6. lines 114-118: I don't understand why to first align at nucleotide level (without apparently relying on codon translation) and then realign at the protein level. Is the manual curation based on nucleotide sequences only? What were the guiding principles? I know you refer to a methodological article, but it would be better to make this one self-contained as you are very close to that objective.

- Methods: Evolutionary histories

7. Good move to start from a publicly available tree (lines 128-129). I am just wondering (1) if all considered species were available in this tree, (2) if you needed to subsample it and (3) on which (presumably single) phylogenetic marker it was originally built. In other words, is its topology fully reliable? This obviously matters for your subsequent analyses.

- Methods: Adaptative evolution

8. lines 137-138: If I understand correctly, subfamily-level alignments were not based on the family-level alignment but realignments from scratch. If correct, please clarify and justify.

9. lines 145-146: I am not sure to understand this. Does this mean that the subsequent analyses were only based on topology (and ignored all branch lengths)? Otherwise, I don't see how you managed to alter the trees. Moreover, the extent of this step is difficult to fathom, especially considering that AQP subfamilies are concerned by multiple duplication events. It looks dangerous.

10. line 152: "null MA" should be expanded on first use.

- Methods: 3D modelling

11. This part is confusing. If I understand correctly (but I am not an expert), you do not need to provide reference sequences to AlphaFold for it to model a given sequence (it is already trained and will fetch homologous sequences automatically). Here, you seem to imply that it used the human sequences as templates, but I think that it was only for surimposition purposes. Moreover, I don't see where you explain the methods behind Figure 4. Finally, the introduction of AlphaFold modelling in the results is too pedantic (lines 583-586). Simply say "we used AlphaFold to model..."

- Results: AQP diversity

12. "clades" (lines 182, 238, 294, 326, 338) suppose rooting, maybe use "groups" or "clans" here. If you disagree with the suggestion, fine, but justify.

13. A general word of caution about public genome completeness and assembly quality could come earlier (e.g. around line 249) and not later on, only about a specific case (lines 299-300).

14. In legend to Figure 2, I don't see the triangles. Besides, what do you mean by a partial sequence? Is there a specific sequence completeness threshold? What about a possible phylogenetic misplacement in these cases?

15. WGD events (R3 and R4) should all be shown on the species trees (e.g. Figures 2 and S2). Otherwise, some arguments are difficult to follow, for example (lines 284-286) how can the additional copies in E. calabaricus and P. senegalus be explained if these species diverged before R3? Are these genes really a and b copies in the sense of the other fishes having experienced R3? Also, in Figure S2, I don't understand why there is no visible contraction in the branch leading to Sparus aurata (only 8 genes).

16. lines 308-310: I am not sure to understand the argument here. The sentence is written in a way that makes the reader think that adaptative evolution has already been addressed whereas you are still discussing presence/absence. I see the motivation for such a transition, but you might want to revise the wording.

- Results: Adaptative evolution

17. lines 320-322: Here I have a somewhat more important point. If I understand correctly, you only tested for positive selection in branches leading to amphibious fishes. In my opinion, you must explain why you did not study this in a completely unbiased way and then showed that, indeed, branches leading to amphibious fishes are more often under positive selection.

18. As for genome completeness above, a word of caution about mis-predicted protein segments should come earlier (e.g. around line 323) and not later on, again only about specific cases (lines 396-398 and 401-402). For this issue, see Di Franco et al. (2019) BMC Evol Biol 19:21 [doi:10.1186/s12862-019-1350-2], where it was proposed to filter sequences beforehand.

19. In Table 1, it is not clear if all omega values are >= 1 (line 353) as soon as the q-value is good enough or if a large omega was required for a branch to be included in the table. Besides, why does omega cannot be above 999? Finally, why the existence of Table S2 with a lone result? Could you not include it in Table 1? Is it because it relies on a different genome sampling?

20. The text describing Table 1 and Figure 3 is sometimes difficult to follow because the former only mention genera or clades, while the text mostly uses abbreviated binomial names. Moreover, I don't understand why they are not 21 sequences in Figure 3. Is Figure 3 limited to the 12 sequences in which specific positions under selection were identified (line 392)? I don't think so, since you refer to Figure 3 (lines 464-466) to say that some sequences were devoid of specific amino-acids under positive selection. This part is quite confusing.

21. lines 385-389: I see your point even if I am not sure that it is the best place to say it. It looks more like an argument to motivate the omega analyses above (around lines 308-310) because there is no AQP subfamily specific to amphibious fishes. That being said, this result was expected as they are a not a monophyletic group and thus are devoid of a basal branch where such gene duplications could have happened (in contrast to tetrapods).

- Results: 3D mapping and modelling

22. I am not sure to understand the point of Figure 4, considering that it is only referred to once and about a point of detail.

23. lines 596-600: I don't understand the reasoning here: both proteins from Danio have the same 7-H fold. Is this additional helix wrongly considered for both genes? How is it possible? Please clarify.

- Supplementary materials

24. The caption of Figure S1 is too long: please remove the sentence beginning with "Branches of amphibious fishes". Moreover, you should explain branch support values and fix the discrepancy (AQP0-15) with Table S1 and the figure itself, where AQP16 do exist.

- Language and typos

lines 70-71: incorrect grammar

line 74: broadly => broad

lines 89-90: repetition of a sentence fragment

lines 153-154: dividing => by dividing

line 256: avoid "on the other hand" twice in a row

line 331: avoid "moreover" twice in a row

line 337: repetition of a sentence fragment

line 340: Our result suggests => Our results suggest

line 347: P .parvicornis => P. parvicornis
